# Resting Energy Expenditure in the Elderly: Systematic Review and Comparison of Equations in an Experimental Population

**DOI:** 10.3390/nu13020458

**Published:** 2021-01-29

**Authors:** Honoria Ocagli, Corrado Lanera, Danila Azzolina, Gianluca Piras, Rozita Soltanmohammadi, Silvia Gallipoli, Claudia Elena Gafare, Monica Cavion, Daniele Roccon, Luca Vedovelli, Giulia Lorenzoni, Dario Gregori

**Affiliations:** 1Unit of Biostatistics, Epidemiology and Public Health, Department of Cardiac, Thoracic, Vascular Sciences, and Public Health, University of Padova, Via Loredan 18, 35121 Padova, Italy; honoria.ocagli@unipd.it (H.O.); corrado.lanera@unipd.it (C.L.); danila.azzolina@uniupo.it (D.A.); gianluca.piras@unipd.it (G.P.); rozita.soltanmohammadi@unipd.it (R.S.); monica.cavion@studenti.unipd.it (M.C.); luca.vedovelli@unipd.it (L.V.); giulia.lorenzoni@unipd.it (G.L.); 2Department of Translational Medicine, University of Piemonte Orientale, Via Solaroli 17, 28100 Novara, Italy; 3ZETA Research Incorporation, Via A. Caccia 8, 34122 Trieste, Italy; silvia.gallipoli@zetaresearch.it; 4Department of Nutrition, University of Buenos Aires and Food and Diet Therapy Service, Acute General Hospital Juan A. Fernandez, Av. Cerviño 3356, Buenos Aires C1425, Argentina; claudiagafare@gmail.com; 5Nursing Home “A. Galvan”, Via Ungheria 340, Pontelongo, 35029 Padova, Italy; direzione@csgalvan.it

**Keywords:** estimating equations, energy requirements, systematic review, elderly, predictive equation, web tool

## Abstract

Elderly patients are at risk of malnutrition and need an appropriate assessment of energy requirements. Predictive equations are widely used to estimate resting energy expenditure (REE). In the study, we conducted a systematic review of REE predictive equations in the elderly population and compared them in an experimental population. Studies involving subjects older than 65 years of age that evaluated the performance of a predictive equation vs. a gold standard were included. The retrieved equations were then tested on a sample of 88 elderly subjects enrolled in an Italian nursing home to evaluate the agreement among the estimated REEs. The agreement was assessed using the intraclass correlation coefficient (ICC). A web application, equationer, was developed to calculate all the estimated REEs according to the available variables. The review identified 68 studies (210 different equations). The agreement among the equations in our sample was higher for equations with fewer parameters, especially those that included body weight, ICC = 0.75 (95% CI = 0.69–0.81). There is great heterogeneity among REE estimates. Such differences should be considered and evaluated when estimates are applied to particularly fragile populations since the results have the potential to impact the patient’s overall clinical outcome.

## 1. Introduction

In the elderly population, malnutrition affects up to 60% of hospitalized patients [1]. Nutritional status, along with aging, is affected by social factors, chronic diseases, physiological changes in body weight, and body composition [2,3]. Malnutrition has multifactorial consequences in older adults. It can lead to a decline in health, with increased episodes of falls [4], vulnerability to infections, and poor wound healing [5]. It also affects functional status, with loss of energy and mobility in daily activities [6].

Moreover, malnutrition affects psychological health by reducing the cognitive state [7], increases morbidity and mortality [8], and has a substantial impact on health care costs [9,10], which is estimated at approximately $15.5 billion in the USA [11]. Quality of life (QoL), the most appropriate endpoint for understanding functional impairments and disabilities, is also compromised in these patients [3,12]. Therefore, to avoid malnutrition and related metabolic stress in frail older adults, the determination of energy needs as part of their daily care is fundamental [13,14].

Daily energy expenditure can vary according to numerous factors, such as age, sex, body composition, clinical condition, and physical activity [15]. Total energy expenditure (TEE) may decrease with aging due to reductions in both the basal metabolic rate (BMR) and physical activity [14], or it can increase due to the rising metabolic turnover and the hypermetabolic effect of fever or medications [16]. Estimated energy requirement (EER), also called energy expenditure estimation (EEE), is an estimation of TEE. BMR, also called basal energy expenditure (BEE), is used to assess energy requirements and contributes to approximately 60–75% of TEE [17]. Since BMR is not easy to measure in daily clinical practice, the resting metabolic rate (RMR), also called resting energy expenditure (REE), is measured [18]. Despite these differences, the terms BMR and RMR are often used interchangeably in the literature [19], as shown in recent reviews [20,21,22]. For simplicity, and as done in previous works, such as ours, we will use the term “resting energy expenditure—REE” in this study and, when appropriate, we will distinguish between the terms.

Indirect calorimetry (IC) [23] and the doubly labeled water method [24] are considered the gold standard methods for estimating energy needs. However, those methods are impractical in daily clinical practice because they are expensive, time-consuming, and require specialized personnel and instrumentation [25]. As a result, several predictive equations have been proposed in the literature to estimate REE. Demographic data (age, sex, ethnicity), anthropometric measurements (height, weight), body composition parameters (fat-free mass, fat mass, organ tissue mass), and, in some cases, specific data (diabetic markers) [17] are the variables most often considered in the proposed equations.

Although they are easy to use, not all the proposed equations may be suitable for each individual, and clinical judgment is still required [15]. Furthermore, the equations have often been validated in a specific population that may have different characteristics than the one being studied [26,27]. The accuracy of these predictive equations is lower in specific populations, such as the elderly population. Validation studies have rarely included older adults, and when included, they were not the main objective of the study [14]. Furthermore, these studies often used variables such as weight (unstandardized for age), which do not adequately explain the change in body composition due to aging.

The purpose of this study is to conduct a systematic review of the REE predictive equations used in the elderly population. We selected only studies that validated their equations against a gold standard (i.e., indirect calorimetry or doubly labeled water). The agreement among the predictive equations retrieved was then evaluated in a sample of elderly patients living in a nursing home. Moreover, to enhance the clinical application of our results, we developed a web application to assist clinicians in choosing the equation that best fits a patient’s available data.

## 2. Materials and Methods

### 2.1. Data Sources

The review followed the Preferred Reporting Items for Systematic Reviews and Meta-analyses (PRISMA) guidelines [28]. We conducted a literature review of MEDLINE (via PubMed), Scopus, and Embase. Table 1 presents a summary of the Population, Intervention or exposure, Comparison, Outcomes, and Study design (PICOS) parameters used to define the inclusion and exclusion criteria for this literature review.

The last update was made on 1 November 2019. The search terms included in the search string were as follows: “energy intake,” “energy intake/physiology,” “basal metabolism,” “nutritional requirements,” “resting metabolic rate,” “resting energy expenditure,” “metabolism,” “energy metabolism” and the additional terms “predictive equations” and “prediction equations.” For the detailed search strategy, see Appendix A.

### 2.2. Eligibility Criteria

#### 2.2.1. Types of Study

Only original studies were included in the review. To be defined as original, the study had to (i) validate a new predictive equation compared to a gold standard method (indirect calorimetry or doubly labeled water method) or (ii) validate an existing equation in a population different than the original ones.

#### 2.2.2. Types of Predictive Equations

To be included in the study, predictive equations (i) must have been based on parameters that are measurable in all possible contexts (i.e., body weight or height), that is to say, they should not require the use of specific equipment; (ii) must include mixed-age patients, at least a portion of whom were over 65 years of age; and (iii) must include equations that are currently used in elderly patients, even if elderly patients were not included in the validation study. Equations based solely on children or adolescents, critically ill patients (burn patients, spinal cord injury patients, patients in a coma, patients who are mechanically ventilated), and people being treated for cancer or chronic kidney injury were excluded because they may have specific nutritional needs.

### 2.3. Data Sources

Figure 1 presents the PRISMA flowchart [28]. The studies were eligible if they had been published in the English, Spanish, or Italian languages, with no limits on the date of publication. Additional sources were sought in the references of all retrieved eligible papers, particularly from reviews.

### 2.4. Data Collection

Two independent reviewers screened the title/abstract/full text of the selected records. Then, full texts were retrieved for further assessment. Each assessor independently extracted information from the eligible studies, such as the use of the equation and the characteristics of the sample in which it was applied. Discrepancies were solved through discussion between the two reviewers in each phase of the review; a third author was consulted when the consensus was not achieved.

### 2.5. Data Extraction

The following key information was extracted from eligible studies and collected in a standard Microsoft Excel sheet: study setting and design, the gold standard used for comparison, study population (number, gender, the presence of disease, body mass index (BMI), age, ethnicity) and predictive equation characteristics (variables in use, agreement with the gold standard). The final data extraction template was modified after reaching consensus in the group based on previous similar work.

### 2.6. Data Synthesis

The characteristics of each study were summarized in the results. Studies were divided according to the inclusion of elderly adults in the validation population.

### 2.7. Predictive Equation Testing

Retrieved equations were tested on a convenience sample of 88 subjects older than 65 years old enrolled prospectively in a nursing home in northern Italy. Data were routinely collected from nursing home medical personnel during routine visits. The administration and the medical personnel approved the study through a collaboration protocol with our department (University of Padova). All procedures were conducted in accordance with the Helsinki Declaration of 1975, as revised in 1983.

Patients receiving enteral or parenteral nutrition and those with edema or ascites, neoplasia, or kidney failure were excluded. For each subject, after oral consent was obtained, a qualified dietitian and a nurse collected anthropometric information and other measurements according to the variables retrieved in the equations obtained from the literature review. All the measurements were taken in the morning between 7 and 10 after overnight fasting. Anthropometric characteristics were measured according to international guidelines using calibrated instruments and previously validated standard protocols [29]. BMI was calculated as weight in kilograms divided by height in meters squared and classified as described by NHLBI consensus [30]. The height in centimeters was measured to the nearest 0.50 cm by a stadiometer. For patients who were unable to stand or were bedridden, knee height was used to estimate height. Weight was measured with the patient in minimal clothing on a digital scale to the closest 0.05 kg after overnight fasting. Skinfold thickness was measured using standard calipers, and the median value of three measurements was considered in the analysis. Ambient temperature and humidity were measured with an electronic hygrometer.

### 2.8. Statistical Methods

Categorical data are reported as relative and absolute frequencies, while continuous data are reported as median and quartiles (I and III). The intraclass correlation coefficient (ICC) was used to evaluate the agreement among the estimated REEs on the convenience sample, with predictive equations as a fixed set of criteria [31]. Equations were grouped as follows for the agreement analysis: (a) equations that consider age; (b) equations that consider gender; (c) equations that consider height; (d) equations that consider weight; (e) equations that consider BMI; (f) equations that consider physical activity; (g) equations that consider more than three variables (three included); (h) equations that include at least one laboratory examination (albumin, glucose level, C reactive protein); (i) equations with at least one measure of the circumference (abdominal circumference, hip circumference, wrist circumference) or that include at least one skinfold measure (chest skinfold, subscapular skinfold); (j) equations including weight and gender; (k) equations with the combination of the variables weight-gender-age; (l) weight-gender-age-height; (m) equations with the combination of the variables weight-gender-age-BMI; and (n) equations with the combination of the variables weight-gender-age-height-BMI equations. For each group, the ICC was determined. The agreement was also determined for BEE-BMR, REE-RMR, and EEE-EER equations since they are representative of different levels of energy requirements. ICC was computed both for the overall sample and for specific subgroups of the sample defined by gender (male/female), obesity (obese/not obese), dysphagia (yes/no), diabetes (yes/no), and Charlson Comorbidity Index (CCI) (≤5, >5). Higher ICCs indicate a higher similarity between values from the same category. The results were reported in forest plots with 95% confidence intervals [CI].

Analyses were performed using R 4.0.2 [32] with the rms [33] and irr [34] packages.

## 3. Results

### 3.1. Literature Review Results

#### 3.1.1. Study Selection

In the initial search, 6353 studies were identified (flowchart in Figure 1). In the final review, 68 studies that developed a new regression equation were included.

The retrieved articles were divided into two groups based on the inclusion of elderly adults in the validation population: in the first group, elderly adults were included (55 equations); in the second group, elderly adults were not included in the original sample but the created equations were subsequently used in this population (Table 2).

The included studies were predominantly cross-sectional in design (27, 39.7%), and 17 were retrospective (25%). The studies were mainly conducted in healthy patients (N = 36, 53%) in an outpatient setting; only 3 studies were carried out in clinical settings [52,60,69]. Only 19 (28%) studies focused on obese patients [39,41,45,46,55,57,58,60,62,65,76,84,86,90,91,92,93,95,99], and 11 (19.6%) studies focused on a diseased population [49,60,79], such as patients with diabetes [51,56,60,82], oncological diseases [60], rheumatoid arthritis [68], chronic obstructive pulmonary disease (COPD) [70], and heart failure [63].

The studies were carried out in Europe, the USA, South America, and Asia. The participants were prevalently Caucasian [37,38,39,43,45,46,48,49,52,53,59,61,62,66,68,71,72,73,74,75,81,84,85,86,88,89,90,94,95,98]; other groups considered were Chinese [40,63,64,67,100], South American [36,41,78,102], Japanese [50,56,58,82,83], Mexican [65,76,93,97,102], African [21,54,69,73], Indian [96] and Australian [99]. Only ten studies were designed exclusively for elderly patients [37,38,47,49,53,66,72,78,79,102].

#### 3.1.2. Energy Expenditure Assessment

In the retrieved studies, indirect calorimetry was the gold standard most frequently used to measure energy expenditure (55 studies, 62.6%). The most common IC instruments applied were respiratory gas analyzer, metabolic cart and open circuit calorimeter; only one study used a wearable device to assess energy expenditure [74]. Some studies compared their results with other previously validated equations as well as with a gold standard. Twenty-six different predictive equations were used as comparisons with the new equations in the articles retrieved: the most frequently used equations were those of Harris Benedict [21,36,37,38,40,41,45,46,48,49,51,56,60,65,68,70,71,72,73,74,76,78,79,80,84,86,89,90,91,92,94,95,98,100], WHO/FAO/UNU, Schofield [36,37,38,57,60,65,74,80,84,89,98,100,102], Owen [37,38,40,45,46,51,62,65,66,68,74,76,78,79,84,86,89,90,91,92,93,95,98,99], Mifflin [37,38,40,45,46,48,51,62,65,68,72,76,78,79,84,86,89,91,92,93,95,98,99], Fredrix [37,38,49,72], Henry [40,60,82,84], Bernstein [51,60,65,84,85,90,91,92,93,95], and Cunningham [65,68,73,89,91,92].

#### 3.1.3. Equation Characteristics

From the literature review, 210 equations were identified. Of these, 13 were validated in a group of patients that did not include elderly adults, and 174 were validated in the elderly population (Figure 1). The variables considered across the equations can be divided as follows (Appendix A): demographic characteristics (age, gender, ethnicity [48,61,98]; menopausal status [37]; smoking [37]; meal status (whether patients had eaten a meal prior to the measurement) [37]); anthropometric measurements (height, weight, BMI [21,49,53,60,71,74,78,82,83,84,94,101]; abdomen [37], hip [51], or wrist circumference [76,101]; arm span [37]; chest skinfold [38] or subscapular skinfold [58]); clinical condition (NYHA [93]; diabetes [55]); physical activity (physical activity [101], leisure time activity [38], athletics [73]); measures of fat percentage (lean body mass [43], surface area [39,81,100]); laboratory tests (glycemia [51,93], albumin [93], C reactive protein [68]); environmental measures (temperature [75,101], humidity [101], time [48]); and vital parameters (body temperature, heart rate and blood pressure [48]). The most commonly used variables were age (147, 70%), gender (166, 79%), weight (183, 87%), and height (86, 41%). BMI was considered in 5 studies (28, 13%).

#### 3.1.4. Precision and Agreement among Equations

Since our review did not evaluate an intervention or a diagnostic tool but instead examined predictive equations, as in Madden’s previous review [22], we did not use the standard Cochrane tools for bias assessment. Stepwise multiple regression was the algorithm most commonly used to select the included variables in the development of predictive equations. Goodness-of-fit was generally assessed in the articles, mainly with the coefficient of determination R^2^, which varied from 0.390 [82] to 0.92 [48]. Only 18 of the 210 equations retrieved were cross-validated or validated in a different sample in the validation study [38,40,41,48,55,56,64,68,70,71,76,86,88,91,92,93,98,100].

### 3.2. Results for the Sample Population

#### 3.2.1. Characteristics of the Sample

The 101 equations were applied to a sample of older adults (27 males and 60 females) living in a nursing home in the Veneto region of Italy. All the equations, except for those that had information that are not available in our sample, were used to compute the REE in our population. For example, the equation of Arciero et al. [38] was not used in our sample since we do not have information regarding leisure time activity. Table 3 presents the descriptive statistics of the sample. The patients had a median age of 74 years, were mostly sedentary (39%, 34) or low activity (18%, 16), had diabetes (75%, 64), and had dysphagia (51%, 44). Appendix A provides the estimated REE for each equation by gender.

#### 3.2.2. Equation Agreement Testing in the Sample Population

Figure 2 reports the ICC of the overall sample, with a higher ICC indicating greater agreement between the estimated REEs. The equations that showed the greatest agreement in the overall population were those that considered laboratory examinations (ICC = 0.81 (95% CI = 0.72–0.87) and weight (ICC = 0.75 (95% CI = 0.70–0.81) in their structure. The equations with the poorest agreement were those that considered BMI and physical activity, with ICCs of 0.43 (95% CI = 0.36–0.52) and 0.23 (95% CI = 0.13–0.35), respectively (Figure 2).

Additionally, in males, equations that included laboratory examinations showed a good agreement level (0.94 [(95% CI = 0.87–0.97)]. In examinations of agreement according to gender, females had a higher overall agreement level of 0.67 (95% CI = 0.59–0.75), with a narrow CI (Figure 3).

For the obese and normal-weight groups, the overall agreement was higher in the obese group, 0.82 (95% CI = 0.72–0.9), and remained high for all the variables considered except for physical activity, which had an ICC of 0.27 for both groups. Equations that included weight in their structure showed higher agreement in dysphagic and diabetic patients and those with a higher Charlson Comorbidity Index (Appendix A). In these groups, the measurement of circumferences agreed well; in contrast, laboratory examinations performed poorly, especially in nondysphagic patients, with an ICC of 0.04 (95% CI = 0.26–0.33), and people with diabetes (0.02 (95% CI = 0.41–0.44)). In all the categories considered, the equations that included physical activity and BMI in their structure had the worst agreement. For the individual agreement (Figure 3), the groups with lower estimated REE had a reduced CI; for example, females had a 1145 Kcal/day estimated REE (95% CI = 1098–1192).

In the forest plots of the CCI, the agreement was lower in both high risk (CCI ≥ 5) and lower risk patients (CCI < 5); equations that considered weight or weight and gender showed greater agreement (Appendix A). Appendix A reprts the agreement among predictive equations in terms of Kcal/die at individual level among the categories gender, BMI, Charlson Comorbidity Index, presence/absence of dysphagia and diabetes. In our sample, in the overall group, BMR and REE had a similar level of agreement, 0.80 to 0.76 respectively for BMR and REE, and EEE had an ICC of approximately 0.46 except when gender was considered (Appendix A).

### 3.3. Web Tool for the Practical Implementation of Equations

In clinical practice, sophisticated instruments such as indirect calorimetry are not always available since they are expensive and require trained personnel [25,103]. This limits their use in daily clinical practice [104]. Furthermore, our results showed that the estimated REE differs according to the equations used. Therefore, a tool is needed to help clinicians estimate REE based on the variables available for the patient. To address this need, we have developed an R Shiny web-based application called *equationer*, which is freely available at the following link https://r-ubesp.dctv.unipd.it/shiny/equationer/. The app is based on the results of this study. The clinicians, after inputting the patient’s available data, will visualize all the estimated REEs based on the equations that considered the variables imputed in their structure. The results will be displayed both graphically (boxplot and bar plot) and tabularly, thus allowing comparisons of the different results of each equation. In the box plot, the app also provides the minimum, maximum, and median values of the estimated REE. For example, the estimated median BMR for a woman with a weight of 65 kg and an age of 75 years is 1249 Kcal/day (min = 1014 Kcal/day, max = 1449 Kcal/day) and 1352 Kcal/day (min = 1225 Kcal/day, max = 1580 Kcal/day), respectively, depending on whether gender is considered in the predictive equations. The median RMR is 1237 Kcal/day (min = 896 Kcal/day and max = 2003 Kcal/day) and 1325 Kcal/day (min = 1188 Kcal/day, max = 1565 Kcal/day), respectively, in equations that do and do not consider gender, and the overall median is 1271 (min 947 Kcal/day and max = 1943 Kcal/day) in equations that consider gender. RMR and BMR have similar median values, and RMR is slightly lower, especially in terms of the minimum value provided, as expected. RMR has great variability, especially in equations that consider gender, and can vary by as much 1107 Kcal/day, whereas BMR ranges are 434 and 355 in equations that do and do not consider gender, respectively. Adding information about physical activity does not increase the median REE value (1241 Kcal/day) (Appendix A). The number of equations resulting from equationer depends on the selected variables. Selecting a choice for categorical variables like, e.g., gender or ethnicity, will result in a lower number of equations estimated. Conversely, setting a value for numerical variables, like, e.g., height or weight, instead will result in a higher number of equations estimated. Detailed instructions on the utilization of the tool are available in the Appendix A.

## 4. Discussion

Given their characteristics of frailty, elderly adults are at risk of malnutrition; hence, it is important to correctly estimate their caloric intake.

This is the first study, to our knowledge, specifically targeting elderly adults, and predictive equations were chosen if (i) they were created for the elderly population, (ii) if elderly subjects were considered in their original sample, and (iii) if they were not included in the validation sample but were widely used in this population. It is worth noting that several reviews already exist on this topic, but none apply our comprehensive inclusion criteria. We have, in fact, extended the criteria to a broader population and considered variables available in clinical practice, such as weight and height. We excluded only variables derived from the use of technological instruments, such as indirect calorimetry. Gaillard [105] included equations with parameters derived from indirect calorimetry; other studies [14,106,107,108] instead consider the equations most frequently used in clinical contexts, do not include elderly adults as a target [21,26,109,110,111], or were addressed to a more specific population [112,113].

Our review shows that a considerable number of predictive energy equations are available in the literature and that they have high variability in the estimated REE when applied in a real sample (Appendix A). This variability could be explained by the fact that the equations were built on a specific population that can have different characteristics from the one in which the equations are used.

Ethnicity has been shown in the literature to influence REE. Our review confirms the results of Compher [114] and shows that this parameter is not widely considered in all equations [115]. However, we were unable to show how differences in ethnicity could affect the estimated REE since our sample included only Caucasians. Equations created for a specific ethnicity, such as for Chinese populations, perform poorly in Caucasian populations, as shown in a recent external validation [20].

The literature reveals that the presence of a specific disease may influence caloric estimation, especially in the elderly. Chronic disease is estimated to affect over 75% of the elderly American population [116] and from 38% to 64% of the Italian population aged from 65 to 69 years, with increased percentages in those over 80 years old [117]. Despite this, in our review, we retrieved only two equations that considered a disease in their structure (diabetes in one case and NYHA classification in the other), even though 11 studies focused on populations with a specific disease.

In the aging population, the physical activity level has a high impact on REE, given the physiological impairments due to aging. Exercise limitations are estimated to increase from 7.7% to 46% in the aging population [117]. In our review, only three equations considered daily living activities. However, adding information about physical activity in our sample worsened the agreement among the equations in the overall sample (ICC = 0.23 (95% CI 0.12–0.35)) (Figure 2) and in all the subgroups considered except patients with diabetes (ICC = 0.48 (95% CI = 0.26–0.70)) (Appendix A). The great variability in physical activity can explain this poor agreement in this population, as can the use of different scores to quantify it.

In our sample, equations that included at least weight or weight and gender yielded a high ICC. In contrast, equations that included variables such as BMI and physical activity had a low agreement in our population for all the considered subgroups.

At the individual level, the agreement was higher in categories that had a lower estimated REE, such as female gender (REE = 1145 Kcal/day (95% CI = 1098–1192)), patients with dysphagia (1173 (95% CI = 115–1231)), normal-weight patients (1179 (95% CI = 1137–1253)) and patients with a Charlson Comorbidity Index higher than 5 (1179 (95% CI = 1115–1231)) (Appendix A). Obese patients have shown high variability in their REE (1364 (95% CI = 1255–1473)), a result in line with those of Bedogni [20], in which equations perform worse with increasing BMI.

Equations that considered the variable age in their structure agreed quite well, from a minimum of 0.57 (95% CI = 0.44–0.7) in male patients and a maximum of 0.79 (95% CI = 0.68–0.90) in obese patients.

BEE-BMR estimation equations agreed better in all the subgroups except for dysphagic patients, where REE-RMR estimation equations were in higher agreement. This could be explained by the fact that the conditions for evaluating BMR were stricter than those for measuring REE in the validation study. Moreover, the EEE-EER equations, which included information on physical activity, showed low agreement in all the subgroups, perhaps because they used different classifications of physical activity.

The female subgroup had a higher level of agreement than the male subgroup. In the example given above for females, the median value changed little (by approximately 100 kcal). At the same time, the minimum and maximum varied up to 1107 when the gender information was included in the structure of the equation. When the same parameter was used for a male person, instead, the median value changed less, and the differences between minimum and maximum for both RMR and BMR were lower, with the highest value produced by equations that included gender (651 Kcal in BMR and 608 in RMR).

The web-based tool derived from this study provides information about the variability of the estimated REEs, which can be viewed easily in the table, the boxplot, and the bar plot. With this information, the clinician can choose the ones most suitable for a patient according to his or her characteristics. The app also provides the minimum, maximum, and median values of REE. At this point, the clinician can choose whether to use the value from a single equation after consulting the original study or to use the median estimated REE, since this seems to be the value that reduces the error best, as shown in previous studies [20,118].

### Limitations

This study does not permit a direct comparison of the retrieved studies due to their substantial differences in the statistical measures used and the different populations considered.

The decision to exclude equations based on body composition parameters could bias the results since fat-free mass is considered a good predictor of REE, especially in elderly people.

The inclusion of equations that had only a minority of older adults in the original sample could reduce the validity of their applicability in older adults, although these equations are used for these populations, and some are even widely used in clinical settings.

The agreement among the equations was evaluated in a small sample with specific characteristics, namely, the prevalence of females and diabetic patients. Therefore, our results are not generalizable to the whole Italian population. It would be useful to repeat the agreement analysis in a large sample and to use a gold standard measure.

Finally, our study considered only Caucasian subjects, although some of the equations were validated in patients of different ethnicities.

## 5. Conclusions

This study provides (i) a relevant examination of the use of predictive equations for elderly adults, (ii) apply the retrieved equations in a convenience sample, and (iii) provide a web application to help the clinician in the choice of the equations to use.

Equations retrieved by this literature review are numerous, consider different variables in their structure, and provide different estimates from one another. Because of the different estimated REEs, that result, choosing one equation over another remains challenging.

The most interesting findings in our work were that in our population, (i) the equations with the highest agreement were those with fewer variables, and (ii) adding information about physical activity and BMI did not increase the agreement among the equations. Since equations with more information reduced the agreement among the equations in our sample, we could suggest avoiding the use of equations that include many variables in their structure, especially for potentially fragile patients, such as those in our sample, for whom all measurements are not usually available. Equations retrieved were usually derived from a specific population; adding variables imply adding coefficient explains the variability of that specific population. This could be the reason why equations with fewer variables showed a higher level of agreement in our population. However, these results must be confirmed by further studies with a broader and more comprehensive sample.

This study was the basis for the development of an easy-to-use tool to guide clinicians in identifying the most appropriate equation for estimating REE based on the subject’s characteristics. The tool allows clinicians to view all the available equations given the characteristics that were entered and to choose the most appropriate equation for the patient. If in doubt, the clinician can use the median value, which is also provided by our tool.

The determination of the exact energy requirements in this population is only the first step in avoiding nutritional problems such as malnutrition and obesity. This vulnerable population requires an overall assessment of nutritional conditions based on the quantification of biomarkers, which is the most objective and unbiased way to assess the intake of particular diet components [119] in addition to appetite evaluation [120] and the use of screening protocols [121].

## Figures and Tables

**Figure 1 nutrients-13-00458-f001:**
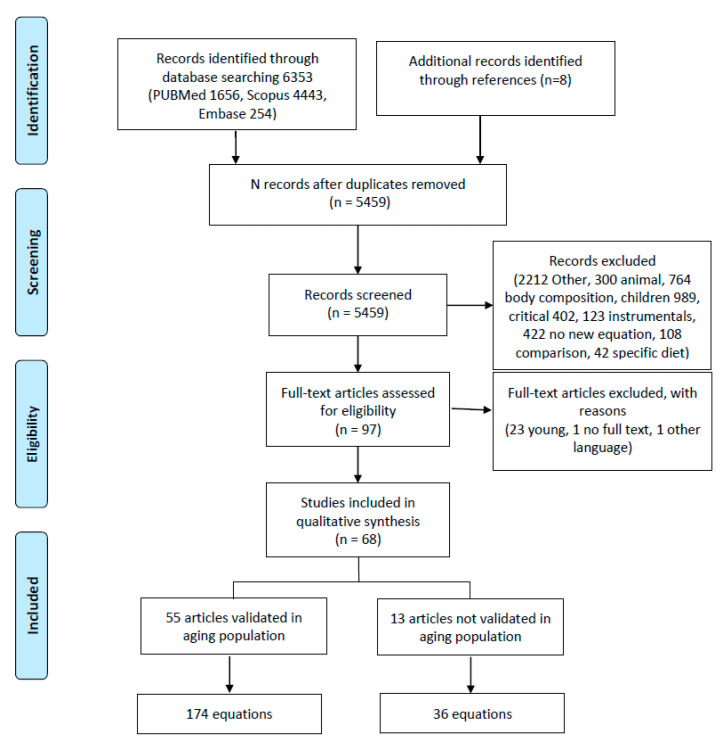
Flowchart of literature search. This figure was based on the PRISMA example.

**Figure 2 nutrients-13-00458-f002:**
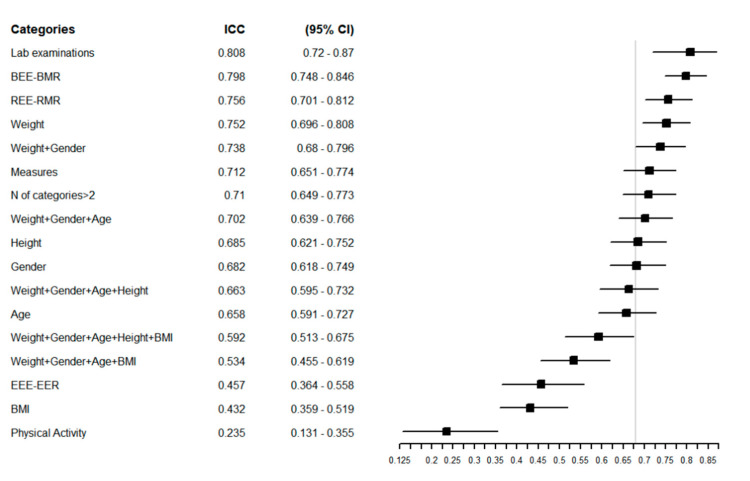
Forest plot reporting the Intraclass Correlation Coefficient (ICC) with 95% CI of estimated REE in the overall population for each specific group of predictive equations. The vertical grey line represents the ICC in the whole category without any grouping: ICC = 0.68 [0.62–0.75] 95% CI.

**Figure 3 nutrients-13-00458-f003:**
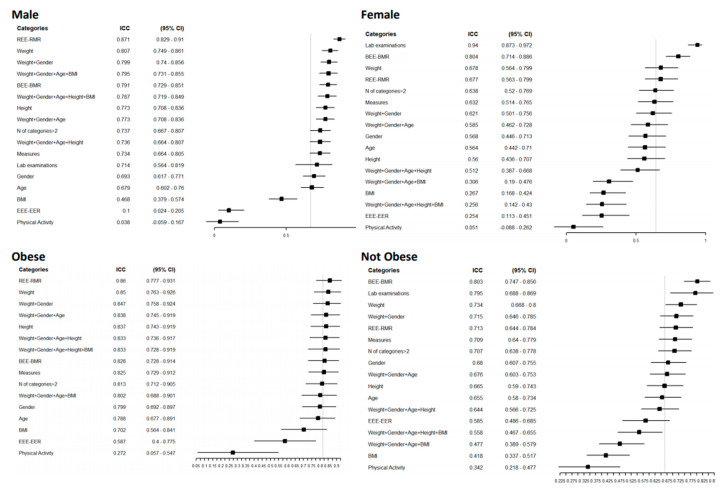
Forest plot reporting the Intraclass Correlation Coefficient (ICC) with 95% CI of estimated REE according to a specific category of patients for a specific group of predictive equations. The vertical grey line represents the ICC in each category without any grouping: Male ICC 0.64 [0.52–0.77] 95% CI, female 0.67 [0.59–0.75], obese 0.82 [0.72–0.90], normal weight 0.67 [0.59–0.74].

**Table 1 nutrients-13-00458-t001:** Summary of Population, Intervention or exposure, Comparison, Outcomes, and Study design (PICOS).

Parameter	Inclusion Criteria	Exclusion Criteria
**Population**	Adult aged >18 years	- Subjects aged < 18 years, - critically ill patients, - people recovering from cancer treatment or in treatment for chronic kidney injury.
**Intervention or exposure**	REE, RMR, BMR, BEE assessed by a brand-new equation	
**Comparison**	Indirect calorimetry, doubly-labeled-water method, or other already validated equations	
**Outcomes**	Predicted caloric intake	
**Study design**	Observational studies	None

**Table 2 nutrients-13-00458-t002:** Predictive equations retrieved by the systematic review. In the table, equations are shown as in the original article (RMR = Resting Metabolic rate, REE = Resting Energy Expenditure, BMR = Basal Metabolic Rate, BEE = Basal Energy Expenditure, 24EE = 24-h energy expenditure). For each equation is reported the formula and characteristics of the population in which is validated and the coefficient of determination (R2). Equations, when not indicated, are expressed in Kcal/day. Continuous variables are reported with mean and standard deviation.

Author	Equation	Study Design	Country	Gold Standard	BMI	Age	N° Patients	Health Status	R^2^
Aleman [35]	M: RMR (MJ/day) = 1.6447 + 0.05714 W + 0.449 (1)	CrS	Cuba, Chile, Mexico	DLW	24.3 ± 4.2	70.1 ± 5.4	19	healthy	0.75
	F: RMR (MJ/day) = 1.6447 + 0.05714 W + 0.449 (0)								
Anjos [36]	M: BMR (KJ/day) = 9.99 W + 7.14 H (m) − 2.79 A − 450.5	CrS	Brazil	IC	15.5–45.3	42.6 SE: 1.4	190	healthy	0.87
	F: BMR (KJ/day) = 8.95 W + 8.87 H (m) − 0.70 A − 814.3				25.4 SE: 0.3	44.9 SE: 1.0	339		0.83
Arciero [37]	F: RMR = 7.8 W + 4.7 H − 39.5 (Menopausal status) + 143.5	CrS	USA	IC	63.3 ± 7	61.8 ± 8	75	healthy	0.59
Arciero [38]	M: RMR = 9.7 W − 6.1 (CS) − 1.8 A + 0.1 LTA + 1060	CrS	USA	IC	77 ± 9	63 ± 8	61		0.76
Bernstein [39]	M: RMR = 11.02 W + 10.23 H − 5.8 A − 1032	CrS	USA	MC	-	40.4 ± 12.6	48	healthy	0.66
	F: RMR = 7.48 W − 0.42 H − 3 A + 844					39.4 ± 12.0	154		0.45
	M: RMR = 1372 BSA + 6.2 A − 1079								0.65
	F: RMR = 758 BSA − 2.3 A − 53								0.42
Camps [40]	M: BMR (kJ/day) = 52.6 W + 828 (1) + 1960	CrS	China	IC	26.9 ± 4.9	21–67	121	healthy	0.81
	F: BMR (kJ/day) = 52.6 W + 828 (0) + 1960	CrS			25.8 ± 5.9	33.4 ± 11.2	111		
Carrasco [41]	F: BMR ≥ 30 A = W 10.9 + 593	CrS	Chile	IC	18.5–69.7	18–74	816	healthy	-
	M: BMR ≥30 A =W 11.2 + 753					18–71	441		
	F, 18–74 A: BMR = W 10.9 − A 2.85 + 716						816		
	M, 18–74 A= W 11.1 − A 2.5 + 864						441		
Cole & Henry [42]	BMR (MJ/day) = exp ^(−0.1614 − 0.00255 A + 0.4721 ln W + 0.2952 ln H)^	CrS	Mixed	-	-	18–80	1207	healthy	-
	M: BMR (MJ/day) = exp ^(−0.2630 − 0.00277 A + 0.4877 * ln W + 0.3367 * ln H)^						6425		
	F: BMR (MJ/day) = e ^(−0.1934 − 0.00199 A + 0.4764 ln W + 0.0194 ln H)^						1030		
	BMR (MJ/day) = exp e ^(−0.0713 − 0.0209 A + 0.4075 ln W + 0.3540 ln H)^						3224		
Cunningham [43]	BMR (cal/day) = 500 + 22 LBM	CrS	USA	IC	59.8 ± 11	29 ± 11	223	healthy	-
	BMR (cal/day) = 601.2 + 21 LBM − 2.6 A								
European Communities [44]	M: BMR (MJ/day), 60 − 74 A = 0.0499 W + 2.93	Re	-	-	Schofield data	-	-	healthy	-
	F: BMR (MJ/day), 60 − 74 A = 0.0386 W + 2.88								
	M: BMR (MJ/day), >75 A = 0.035 W + 3.43								
	F: BMR (MJ/day), >75 A = 0.0410 W + 2.61								
Frankenfield [45]	M: O: RMR = W 10 + H 3 − A 5 + 244 + 440	CrS	USA	IC	18.6 ± 1.5	18–85	337	healthy	0.84
	M: NW: RMR = W 10 + H 3 − A 5 + 207 + 454							obese	
	M: O: RMR = W 10 − A 5 + 274 + 865								
	M: NW: RMR = W 11 − A 6 + 230 + 838								
Frankenfield [46]	M: RMR = 66 + 13.75 W + 5.0 H − 6.76 A	CrS	USA	IC	18.8–96.8	18–78	54	diabetic	-
	F: RMR = 655 + 9.56 W + 1.85 H − 4.68 A						76		
Fredrix [47]	M: REE = 1641 + 10.7 W − 9.0 A − 203 (1)	CrS	Netherlands	IC	25.5 ± 2.6	51–82	18	healthy	0.92
	F: REE = 1641 + 10.7 W − 9.0 A − 203 (2)				26.4 ± 2.4	66 ± 7	22		
Freni [48]	M: RMR = 635.8 + 12.98 W	CrS	USA	IC	-	25–74	76	healthy	0.61
	M: RMR = 1007.5 + 12.48 W − 7.84A								0.7
	M: RMR = 1002.8 + 12.15 W − 7.35 A + 154.56 smoke								0.71
	M: RMR = 687.2 + 11.08 W − 6.84 A + 162.00 smoke + 7.48 bpdif								0.76
	M: RMR = 1138.2 + 11.44 W − 7.13 A + 228.62 smoke + 5.79 bpdif + 137.93 race − 67.85 T + 163.92 3 meal								0.81
	F: RMR = 681.5 + 9.16 W								0.58
	F: RMR = 785.2 + 9.36 W − 2.48 A								0.6
	F: RMR = 771.1 + 9.95 W − 2.58 A + 110.54 smoke								0.62
	F: RMR = 711.4 + 9.15 W − 3.88 A + 112.56 smoke + 3.07 bpdif								0.64
	F: RMR = −1492.0 + 9.58 W − 3.55 A + 81.00 smoke + 1.94 bpdif + 78.31 race + 4.19 pulse + 51.93 BT								0.71
Gaillard [49]	Eq1: BMI > 21, REE = 18.84 W	CrS	France	IC	25.2 ± 5.5	80.7 ± 8.6	187 (60)	diseased	0.005
	Eq1: BMI ≤ 21, REE = 22.29 W								0.006
	Eq2: REE = 82.6 + 9.5 W + 6.5 H − 6.1 A								0.164
	Eq3: REE = 497 + 11.6 W								0.232
Ganpule [50]	M: RMR = 0.0481 W + 0.0234 H − 0.0138 A − 0.5473 (0) + 0.1238	CrS	Japan	IC	23.4 ± 3.1	36 ± 16	71	healthy	0.834
	F: RMR = 0.0481 W + 0.0234 H − 0.0138 A − 0.5473 (1) + 0.1238				21.4 ± 3.3	37 ± 16	66		
Gougeon [51]	REE (KJ/day) = 4044 + 79 W + 78 FPG − 43 HC	CrS		IC	37 ± 1	54 ± 2	25	healthy	0.813
Harris & Benedict [52]	F: RMR = 1.8496 H + 9.5634 W − 4.6756 A + 655.0955	CrS	USA	IC	-	29 ± 14	103	healthy	0.59
	M: RMR = 66.4730 + 13.7516 W + 5.0033 H − 6.7550 A						136		
Hedayati & Dittmar [53]	M: REE = 41.567 − 0.226 AC	CrS	Germany	IC	26.0 ± 2.67	68.4 ± 4.48	51	healthy	-
	F: REE = 46.155 − 0.273 HC				25.0 ± 3.29	68.1 ± 5.15	49		
	F: REE = 69.865 − 0.229 HC − 0.173 H (m)				25.0 ± 3.29	68.1 ± 5.15	49		
	F: REE = 68.143 − 0.025 HC − 0.210 H (m) − 0.519 BMI				25.0 ± 3.29	68.1 ± 5.15	49		
Henry [54]	M > 60 A: BMR = 13.5 W +514	Re	Mixed	IC	-	-	534	healthy	-
	F > 60 A: BMR = 10.1 W + 569						334		
	M 60–70 A: 13.0 W + 567						270		
	M > 70 A: BMR = 13.7 W + 481						264		
	F 60–70 A: BMR = 10.2 W + 572						185		
	F > 70 A: BMR = 10.0 + 577						155		
Huang [55]	M, O, diabetic: RMR = 71.767 − 2.337 A + 257.293 (1) + 9.996 W + 4.132 H + 145.959 DM (1)	-	Australia	IC	48.0 ± 7.9	51.9 ± 11.7	61	healthy	0.75
	M, O, non-diabetic: RMR = 71.767 − 2.337 A + 257.293 (1) + 9.996 W + 4.132 H + 145.959 DM (0)				47.1 ± 9.2	43.9 ± 12.9	218		
	F, O, diabetic: RMR = 71.767 − 2.337 A + 257.293 (0) + 9.996 W + 4.132 H + 145.959 DM (1)				47.4 ± 8.8	51.6 ± 11.9	81		
	F, O, non-diabetic: RMR = 71.767 − 2.337 A + 257.293 (0) + 9.996 W + 4.132 H + 145.959 DM (0)				46.0 ± 8.2	43.7 ± 12.4	678		
Ikeda [56]	M: BEE = 10 W − 3 A + 125 (1) + 750	P	Japan	IC	23.9 ± 5.3	58.3 ± 10.3	39	healthy	0.81
	F: BEE = 10 W − 3 A + 125 (0) + 750				24.2 ± 3.8	61.8 ± 12.2	29		
Institute of Medicine (U.S.) [57]	M: BEE (NW, OW, O) = 293 − 3.8 A + 456.4 H (m) + 10.12 W	-	-	DLW	-	-	-	-	0.64
	F: BEE (NW, OW, O) = 247 − 2.67 A + 401.5 H (m) + 8.6 W								0.62
	M: BEE (NW) = 204 − 4 A + 450.5 H (m) + 11.69 W								0.46
	F: BEE (NW) = 255 − 2.35 A + 361.6 H (m) + 9.39 W								0.39
	M, 18.5 ≤ BMI ≤ 25, EER = 661.8 − 9.53 A + PAL 15.91 W + 539.6 H (m)								
	F, 18.5 ≤ BMI ≤ 25, EER = 354.1 − 6.91 A + PAL 9.36 W + 726 H (m)								
	M, BMI > 25, EER = 1085.6 − 10.08 A + PAL 13.7 W + 416 H (m)								
	F, BMI > 25, EER = 447.6 − 7.95 A + PAL 11.4 W + 619 H (m)								
Kashiwazaki [58]	RMR = 22.7 W − 13.6 SSF + 350.6	P	Japan	IC	23.6 ± 3.1	36.5 ± 10.4	134 (66)	healthy	-
Korth [59]	REE (kJ/day) = 65.6 W + 2284	CrS	Germany	IC	-	-	-	healthy	0.46
	M: REE (kJ/day) = 41.5 W − 19.1 A + 35.0 H + 1107.4 (1) − 1731.2				25.9 ± 7.4	37.1 15.1	50		0.71
	F: REE (kJ/day) = 41.5 W − 19.1 A + 35.0 H + 1107.4 (0) − 1731.2				25.5 ± 4.4	35.3 ± 15.4	54		
Kruizenga [60]	M: BMI < 25: REE = 11.355 W + 7.224 H − 4.649 A + 135.265 (1) − 137.475	P	Netherlands	IC	23.4 ± 7.2	53 ± 15.6	260	diseased	-
	F: BMI < 25: REE = 11.355 W + 7.224 H − 4.649 A + 135.265 (0) − 137.475						253		
Lam [61]	M, AA: 24EE = 11.6 W + 8.03 H − 3.45 A + 217 (1) − 52 (1) − 235	Re	USA	IC	29.3 ± 7.0	34.5 ± 11.9	211	healthy	0.797
	M, wh: 24EE = 11.6 W + 8.03 H − 3.45 A + 217 (1) − 52 (0) − 235						211		
	F, AA: 24EE = 11.6 W + 8.03 H − 3.45 A + 217 (0) − 52 (1) − 235						270		
	F, wh. 24EE = 11.6 W + 8.03 H − 3.45 A + 217 (0) − 52 (0) − 235						270		
Lazzer [62]	M: BMR (kJ/day) = 46 W − 14 A + 1140 (1) + 3252	P	Italy	IC	41.6 ± 6.8	46.3 ± 13.8	2000	healthy	0.6
	F: BMR (kj/day) = 46 W − 14 A + 1140 (0) + 3252				41.9 ± 6.5	47.8 ± 13.9	5368		
Leung [63]	REE (KJ/day): 57.562 W − 26.795 A + 3340.2	P	China	IC	23.6 ± 3.8, 23.1 ± 4.1	45 ± 17, 72 ± −10	70	healthy	0.619
Liu [64]	M: BMR = 13.88 W + 4.16 H − 3.43 A − 112.40 (0) + 54.34	CrS	China	IC	22.6 ± 2.4	44 ± 15.0	102	healthy	0.81
	F: BMR = 13.88 W + 4.16 H − 3.43 A − 112.40 (1) + 54.34				21.5 ± 2.2	43.6 ± 13.7	121		0.81
	BMR = 20.29 W + 29.34								0.65
	BMR = 13.51 W + 11.93 H − 1506.60								0.75
	M: BMR = 14.73 W − 3.87 A − 150.90 (0) + 755.30								0.8
	F: BMR = 14.73 W − 3.87 A − 150.90 (1) + 755.30								0.8
Livingston & Kohlstadt [65]	F: RMR = 248 W^0.4356^ − 5.09 A	R	USA	IC	-	-	-	healthy	0.67
	M: RMR = 293 W ^0.4330^ − 5.92 A								0.73
	F: RMR = 196 W ^0.4613^								0.67
	M: RMR = 246 W ^0.4473^								0.73
	RMR = 202 W ^0.4722^								0.64
	RMR = 261 W ^0.4456^ − 6.52 A								0.68
Lührmann [66]	M: RMR (kJ/day) = 3169 + 50.0 W − 15.3 A + 746 (1)	Lo	Germany	IC	26.3 ± 3.1	66.9 ± 5.2	107	healthy	0.74
	F: RMR (kJ/day) = 3169 + 50.0 W − 15.3 A + 746 (0)				26.4 ± 3.7	67.8 ± 5.7	179		
	RMR (kJ/day) =1238 + 66.4 W								0.62
	F: RMR (kJ/day) = 2078 + 50.8 W + 751 (0)								0.73
	M: RMR (kJ/day) = 2078 + 50.8 W + 751 (1)								
Lv [67]	M: EER (MJ/day) = −0.030 A + 0.287 (1) + 0.131 H − 0.104 W − 0.031 WC + 0.263 PL − 5.172	CT	China	IC	27.16 ± 3.45	54 ± 7	135	healthy	-
	F: EER (MJ/day) = −0.030 A + 0.287 (0) + 0.131 H − 0.104 W − 0.031 WC + 0.263 PL − 5.172						81		
Metsios [68]	REE = 598.8 W ^0.47^ A ^−0.29^ CRP^0.066^	R	United Kingdom	IC	26.2 ± 5.6	62.0 ± 10.2	82	Rheumatoid arthritis	0.62
Mifflin [69]	M: RMR = 9.99 W + 6.25 H − 4.92 A + 166 (1) − 161	Obs	Mixed americans	IC	27.5 ± 4.1	44.4 ± 14.3	251	healthy	0.71
	F: RMR = 9.99 W + 6.25 H − 4.92 A + 166 (0) − 161				26.2 ± 4.9	44.6 ± 12.7	247		
	REE = 15.1 W + 371								0.56
	M: REE = 12.3 W + 704								0.36
	F: REE = 10.9 W + 586								0.5
	F: REE (kJ) = 282.630 + (−15.124 A) + 24.481 H + 31.870 W + 243.226 (1)								
Moore & Angelillo [70]	M: REE = 11.5 W + 952	P	USA	IC	-	-	93	COPD	-
	F: REE = 14.1 W + 515						31		
Müller [71]	M: REE (MJ/day) = 0.047 W + 1.009 (1) − 0.01452 A + 3.21	Re	Germany	IC	27.1 ± 7.7	44.2 ± 17.3	388	-	0.73
	F: REE (MJ/day) = 0.047 W + 1.009 (0) − 0.01452 A + 3.21						658		
	M, BMI ≤ 18.5: REE (MJ/day) = 0.07122 W − 0.02149 A + 0.82 (1) + 0.731								
	F: BMI ≤ 18.5: REE (MJ/day) = 0.07122 W − 0.02149 A + 0.82 (0) + 0.731								
	M, BMI > 18.5–25: REE (MJ/day) = 0.02219 W + 0.02118 H + 0.884 (1) − 0.01191 A + 1.233								
	F, BMI > 18.5–25: REE (MJ/day) = 0.02219 W + 0.02118 H + 0.884 (0) − 0.01191 A + 1.233								
	M, 25 < BMI < 30: REE (MJ/day) = 0.04507 W + 1.006 (1) − 0.01553 A + 3.407								
	F, 25 < BMI < 30: REE (MJ/day) = 0.04507 W + 1.006 (0) − 0.01553 A + 3.407								
	M, BMI ≥30: REE (MJ/day) = 0.05 W + 1.103 (1) − 0.01586 A + 2.924								
	F, BMI ≥ 30: REE (MJ/day) = 0.05 W + 1.103 (0) − 0.01586 A + 2.924								
	F: REE (MJ/day) = 0.047 W + 1.009 (0) − 0.01452 H + 3.21								0.73
Obisesan [72]	RMR = 12.2 W + 1.6 FPG (gm/dL) + 103 (NYHA; III, IV) − 144 (albumin mg/dL) + 755	P	Mixed	IC	25.4 ± 5.5	70 ± 7	-	-	0.83
Owen [73]	F, 18–65 A, athletic: RMR = 50.4 + 21.1 W	P	Mixed	-	-	18–56	-	heart failure	-
	M, non-athletic: RMR = 879 + 10.2 W		Mixed		28.2 ± 7.5	38 ± 15.6	60		
	F: nonathletic: RMR = 795 + 7.18 W				20–59	18–65	44		
Pavlidou [74]	M: RMR = 25.41 BMI ^(−0.2115)^	CT	Greece	fitmate	32.0 ± 6.9	10–77	105	-	-
	F: RMR = 21.09 BMI ^(−0.1786)^				29.8 ± 7.6	12–76	278		
	RMR = 21.53 BMI^−0.152^								
Quenouille [75]	BMR = 2.975 H + 8.90 W + 11.7 BSA + 3.0 h − 4.0 AT + 293.8	S	Northern Europe	-	-	-	-	-	-
Quiroz-Alguin [76]	M: REE = 12.204 W − 244.892 (0) + 83.954 WrC − 402.204	P	Mexico	IC	34.7 ± 5.7	18–70	38	obese	0.52
	F: REE = 12.204 W − 244.892 (1) + 83.954 WrC − 402.204						39		
Sabounchi [21]	BMR = 301 + 10.2W + 3.09 H − 3.09 A	Me	Mixed	IC	-	-	-	obese	-
Schofield [77]	M, ≥60 A: REE = 11.711 W + 587.7								
	F, ≥ 60 A: REE= 9.082 W + 658.5								
Segura-Badilla [78]	Eq1, F: REE = 11.701 W + 5.75 H − 7.824 A − 35.95	CrS	Chile	IC	28.0 ± 4.9	67.6 ± 4.5	50	-	0.673
	Eq1, M: 346.867 + 4.317 W + 7.967 H − 10.16 A				28.1 ± 3.1	68.2 ± 4.0	13		
	Eq2, F: REE = 11.774 W + 7.37 H − 817.918								0.649
	Eq2, M: REE = 4.255 W +7.819 H − 316.398								
	Eq3, F: REE = 9427.775 + 84.689 W − 55.063 H − 174.811 BMI − 8.798 A								0.711
	Eq3, M: REE = 41.687 H + 95.416 BMI − 13.978 A − 30.019 W − 5008.038								
	Eq4, F, NW: REE = 896.249 + 14.361 W − 0.055 H − 10.389 A								0.733
	Eq4, F, OW: REE = 17.211 W + 4.437 H − 7.499 A − 314.07								
	Eq4, M, NW: REE = 151.717 H + 24.108 A − 137.022 W − 15817.35								
	Eq4, F, OW: REE = 19.995 + 3.252 W + 9.488 H − 7.61 A								
Silver [79]	REE = 21–23 W	Re	USA	IC	23.0 ± 4.0	86.1 ± 7.3	10	cognitive impairment	-
Sridhar [80]	REE (MJ/day) = 0.295 MAMC + 0.0483 AS − 0.0324 A − 6.25	-	-	IC	-	59.6 ± 8.8	20 (5)	musculoskeletal deformities	0.861
	REE (MJ/day) = 2.38 + 0.0553 W								0.702
	REE (MJ/day) = 0.0554 W + 4.1 − 0.029 A								0.745
	REE (MJ/day) = 0.0436 W + 0.0304 AS − 0.0275 A − 0.26								0.804
	REE (MJ/day) = 0.0102 W + 0.0427 AS + 0241 MAMC − 0.0318 A − 4.88								0.856
	REE (MJ/day) = 0.399 MAMC − 2.27								0.694
	REE (MJ/day) = 0.393 MAMC − 0.0247 A − 170								0.714
Staats [81]	M: BCR (Kcal/h) = (43.66 − 0.1329 A) BSA	Re	Germany	-	-	20–74	639	diabete	-
	F: BCR (Kcal/h) = (38.65 − 0.0909 A) BSA						828		
Tabata [82]	M, 50–69 A, BMR = 21.5 (65.0)	Re	Japan	DLW	22.7 ± 2.9	39 ± 10	-	healthy	-
	F, 50–69 A, BMR = 20.7 (53.6)								
	M, ≥70 A: BMR = 21.5 (59.7)								
	F, ≥70 A: BMR = 20.7 (49.0)								
Tabata [83]	BMR = 797 + 15.7 W − 8.30A	-	Japan	-	25.7 ± 4.1	60 ± 12	69	diabetes	0.67
	M: BMR = 957 − 11.6 A + 38.5 BMI + 200 (1)				25.7 ± 4.1	57 ± 12	37		0.67
	F: BMR = 957 − 11.6 A + 38.5 BMI + 200 (0)				26.1 ± 3.4	64 ± 11	32		
Weijs [84]	M: BMI > 25: REE = 14.038 W + 4.498 H − 0.977 A + 137.566 (1) − 221.631	P	Belgium, Germany	IC	35.2 ± 7.7	18–71	95	obese	0.69
	F: BMI > 25: REE = 14.038 W + 4.498 H − 0.977 A + 137.566 (0) − 221.631						41		0.69
WHO [85]	M > 60 A: BMR = 8.8 W + 1128 H − 1071								
	F > 60 A: RMR = 9.2 W + 637 H − 302								
Wilms [86]	F: REE = 816.714 + 11.035 W − 3.435 A	P	Germany	IC	42.8 ± 7.0	41.7 ± 13.2	273	obese	0.57
Xue [87]	M: RMR = 13.9 W + 247 (1) − 5.39 A +855	CrS	China	IC	16.7–38.2	18–67	315	healthy	0.607
	F: RMR = 13.9 W + 247 (0) − 5.39 A +855								
**Equations validated in a population aged lower than 65**									
De la Cruz Marcos [88]	M: REE = 1 376.4 − 308 (0) + 11.1 W − 8 A	CrS	Spain	IC	22.2 ± 1.9	19–65	45	healthy	0.68
	F: REE = 1 376.4 − 308 (1) + 11.1 W − 8 A	CrS					50		
De Lorenzo [89]	M: RMR (kJ/day) = 53.284 W + 20.957 H − 23.859 A + 487	CrS	Italy	IC	26.7 ± 4.3	28.7 ± 11.4	127	healthy	0.597
	F: RMR (kJ/day) = 46.322 W + 15.744 H − 16.66 A + 944	CrS			27.8 ± 5.1	41 ± 11.5	193		0.597
de Luis [90]	M: REE = 58.6 + 6.1 W + 1023.7 H (m) − 9.5 A	CrS	Spain	IC	35.6 ± 5.7	43.7 ± 15.3	60	obese	-
	F: REE = 1272.5 + 9.8 W − 61.6 H (m) − 8.2 A	CrS			34.9 ± 5.2	46.6 ± 17.5	140		
Lazzer [91]	M: REE (MJ/day) = 0.048 W + 4.655 H − 0.020 A − 3.605	P	Italy	IC	45.4	20–65	164	obese	0.68
Lazzer [92]	F: REE (MJ/day) = 0.042 W + 3.619 H − 2.678	P	Italy	IC	45.6	19–60	182	obese	0.66
Orozco-Ruiz [93]	F: REE = 12.114 W − 6.541 A + 835.952	CrS	Mexico	IC	31.4 ± 4.34	39.1 ± 10.9	303	obese	0.51
	M: REE = 12.114 W − 6.541 A + 1094.991						107		
Roza & Shizgal [94]	M: RMR = 88.362 + 4.799 H + 13.397 W − 5.677 A	Re	USA	IC	-	30 ± 14	168	healthy	-
	F: RMR = 447.593 + 3.098 H + 9.247 W − 4.330 A					31 ± 14	169		
	M: RMR = 77.607 + 4.923H + 13.702 W − 6.673 A								
	F: RMR = 667.051 + 1.729 H + 9.74 W − 4.737 A								
	M: RMR = 75.9 + 1.3 A + 53.7BMI								
	F: RMR = 490.8 − 1.5A + 45.8 BMI								
Siervo [95]	F: REE = 542.2 + 11.5 W	P	Italy	IC	31.81 ± 4.97	23.78 ± 3.79	157	obese	0.59
Soares [96]	BMR (kj/day) = 48.7 W − 14.1 A + 3599	P	Indian	IC			121	healthy	
Valencia, & Haggarty [97]	F: BMR = 10.98 W + 520	P	Mexico	IC		18–40		healthy	
	M: BMR = 14.21 W + 42						32		
Vander Weg [98]	F: AA: REE = 147.45 − 3.56 A + 8.39 W + 4.74 H − 64.98 (1)	CrS	USA	IC	25.2	18–39	239	healthy	0.51
	F: wh: REE = 147.45 − 3.56 A + 8.39 W + 4.74 H − 64.98 (0)					18–37			
Wright [99]	M: RMR = 9.27 W + 4.58 H − 6.53 A + 451.44	R	Australia	IC	32.0 ± 5.6	46.4 ± 10.4	154	obese	-
	F: RMR = 9.02 W + 5.88 H − 7.47 A + 110.76				32.9 ± 5.8	47.4 ± 11.0	124		
	M, OW: RMR = 2.91 W − 1.83 H − 11.12 A + 2372.11								
	F, OW: RMR = − 4.28 W + 20.17 H − 7.50 A − 1295.89								
	M, O: RMR = 9.19 W + 12.96 H − 2.34 A − 1233.82								
	F, O: RMR = 7.23 W + 6.83 H − 6.78 A + 113.90								
Yang [100]	M: BEE (kJ/day) = 277 + 89 W + 600 (1)	P	China	IC	21.03 ± 0.17	30.66 ± 0.94	79	healthy	0.48
	F: BEE (kJ/day) = 277 + 89 W + 600 (0)				20.81 ± 0.18	31.01 ± 0.87	86		
	BEE (kJ/day) = 6285 BSA − 4611								0.5
	BEE (kJ/day) = 103 H − 11189								0.45
	BEE (kJ/day) = 114 W − 801								0.44
	M: BEE (kJ/day) = 105 W − 58								0.27
	F: BEE (kJ/day) = 69 W + 1355								0.24
Yangmei [101]	M: ER (MJ/day) = 13.5 − 0.025 A + 0.215 AI − 0.006 WC + 0.342 AT − 0.268 BMI + 0.623 (1)	P	China	IC	27.40 ± 2.34	53 ± 21(F)	1292	Metabolic syndrome	-
	F: ER (MJ/day) = 13.5 − 0.025 A + 0.215 AI − 0.006 WC + 0.342 AT − 0.268 BMI + 0.623 (0)								

General abbreviations: A = age (years), BMI = body mass index, BMR = basal metabolic rate, BEE = Basal energy expenditure, DLW = Doubly Labelled Water, CrS = Cross-sectional, CT = clinical trial, F = female, H = height (cm), EEE = energy expenditure estimation, EER = estimated energy requirement, IC = Indirect Calorimetry, Lo = Longitudinal, M = male, MC = Metabolic chart, Me = Metaregression, NW = normal weight, O = Obese, Obs = Observational, P = Prospective, OW = overweight, R = retrospective, Re = Reanalysis, REE = resting energy expenditure, RMR = resting metabolic rate, S = survey, W = weight (kg). Abbreviations among equations: AA = African American, AC = Abdomen Circumference (cm), AS = arm span (cm), AT = ambient temperature, CRP =C Reactive Protein (mg/L), CS = Chest skinfold (mm), BT = body temperature (°C), DM = diabetes mellitus (1 = yes, 0 = no), FPG = Fasting plasma glucose (mmol), h = humidity, HC = Hip circumference (cm), MAC = midarm circumference (cm), MAMC = midarm muscle circumference (cm) MAMC = MAC − 3.14 TSF (Triceps skinfold thickness mm), SSF = subscapular skinfold (mm), T = hour (decimalized hour of day that RMR was measured (range 7.8–12.1)), WC = waist circumference, wh = white, wrc= wrist circumference (cm). Levels of variables: AI = activity Intensity Index (0, low physical job; 1, medium physical job) (Yangmei); AT = ambient temperature in Yangmei (0, 10–308 °C; 1, <10 °C or >30 °C); Bpdif: blood pressure gradient (systolic-diastolic) (mmhg); Meal: 0 = fasting, 1 = for having had breakfast prior to calorimetry; smoke: 0 = current non-smokers, 1 = current smokers; race: 0 = black, 1 = white; BSA = Body Surface Area (BSA = Body Surface Area (0.007184 H 0.725 * W0.425, Dubois& Du Bois 1916); LBM = Lean Body Mass (M: LBM = (79.5 − 0.24 W − 0.15 A) W/73.2; F: LBM = (69.8 − 0.26 W − 0.12 A) W/73.2 (Moore et al., 1963)); LTA = Leisure Time Activity (see the Minnesota Leisure Time Physical Activity questionnaire, Taylor et al., 1978) (Arciero), PAL = Physical Activity Level: PA = 1.00 if PAL is estimated to be ≥1.0 <1.4 (sedentary), PA = 1.13 if PAL is estimated to be ≥1.4 <1.6 (low active), PA = 1.26 if PAL is estimated to be ≥1.6 <1.9 (active), PA = 1.42 if PAL is estimated to be ≥1.9 <2.5 (very active) (IOM); Menopausal Status: 1 = perimenopausal women, 2 = perimenopausal women (vasomotor instability, “hot flashes”, absence of regular menstruation for 2 to 12 months), 3 = post-menopausal women (absence of menstruation for greater than 12 months); NYHA = New York Heart Association (I, No limitation of physical activity. Ordinary physical activity does not cause undue fatigue, palpitation, dyspnoea (shortness of breath); II, Slight limitation of physical activity. Comfortable at rest. Ordinary physical activity results in fatigue, palpitation, dyspnoea (shortness of breath). III, Marked limitation of physical activity. Comfortable at rest. Less than ordinary activity causes fatigue, palpitation, or dyspnoea. IV, Unable to carry on any physical activity without discomfort. Symptoms of heart failure at rest. If any physical activity is undertaken, discomfort increases.

**Table 3 nutrients-13-00458-t003:** Descriptive characteristics of the sample. Categorical data are reported as relative and absolute frequencies; continuous data as median, I, and III quartiles.

Variable	Level	*N*	Statistics
Anthropometric characteristics			
Age		87	74.0/83.0/90.0
Gender	Female	87	68% (60)
Ethnicity	Caucasian	87	100% (87)
Menopausal Status	pre	60	3% (2)
	peri		13% (8)
	post		83% (50)
Measurements			
Mean Chest Skinfold		66	10.0/13.5/17.0
Mean Subscapular Skinfold		60	13.0/16.1/19.1
Waist Circumference		44	86.8/95.8/103.0
Wrist Circumference		74	15.0/16.0/17.0
Arm Circumference		73	23.0/26.0/28.4
Weight (Kg)		86	51.6/62.0/69.7
Height (cm)		74	144/151/157
Clinical condition			
Diabetes	yes	85	75% (64)
Dysphagia	yes	87	51% (44)
Fall Risk	yes	86	1% (1)
Hospital Admission	yes	59	76% (45)
Charlson Comorbidity Index		87	4/5/6
Parkinson/Alzheimer	yes	13	46% (6)
Blood examinations	Glycemia	47	79.0/92.0/101.0
	Urea (mmol/L)	41	5.00/7.00/9.90
	Creatinine (umol/L)	55	69.5/83.0/133.0
	C Reactive Protein	17	2.52/3.86/11.83
Physical activity			
Physical Activity (IOM)	Sedentary	87	39% (34)
	Low Active	-	18% (16)
	Active	-	39% (34)
	Very Active	-	3% (3)

## Data Availability

The data presented in this study are available on request from the corresponding author.

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
