# Peer review of "Resting Energy Expenditure in the Elderly: Systematic Review and Comparison of Equations in an Experimental Population"

_nutrients, 2021, doi:10.3390/nu13020458_

Round 1

Reviewer 1 Report

Review of “Resting energy expenditure in the elderly: systematic review 2 and comparison of equations in an experimental population”

First, let me congratulate you on a clever approach to cutting through the fog regarding which equation for BMR/RMR is “best”. The addition of the online tool to translate the research findings to clinical tools was truly creative.

That said, I think that some revisions are needed to make the paper stronger.

First, I do not believe that you carried out a systematic review (which includes an assessment of risk of bias), but rather a scoping review. See https://bmcmedresmethodol.biomedcentral.com/articles/10.1186/s12874-018-0611-x . With the difference between a scoping and a systematic review cleared up, then statements like the one on line 245 about the Cochrane tool are not needed. Thus, I think you need to reframe what you call the first part of the analysis (and, hence, the title of the paper).

With regard to the review component of this study:

  • The flow chart is confusing: it appears that there were two title/abstract screens. Please clarify what differentiates the two phases of screening.
  • The next to last box bottom right has been truncated.
  • Please provide a bit more detail regarding the data extraction template. I'm assuming this was a customized template, but was it initially screened for use by the team? Any modifications?
  • On line 127, I think you mean "characteristics of the sample" rather than "population".
  • Section 2.5 Data Synthesis: This section really belongs in 2.4 Data Collection as it lists the specific pieces of information extracted from the articles. Typically, the data synthesis section would describe the meta-analytic models used, methods for imputation or data conversion, subgroup analyses or meta-regressions to explain heterogeneity. Or, if no quantitative synthesis was carried out the use of particular conceptual or theoretical frameworks that guided the qualitative interpretation. That said, it doesn't appear that any particular synthesis was carried out as it seems that the goal of the systematic (actually scoping) review was to identify the equations rather than attempt a traditional synthesis. The lack of typical synthesis for a systematic review is not a flaw, but it should be made clear to the reader who might be expecting something different.

With regard to the observational agreement study:

  • Line 137: you describe your sample as a “casual” sample, though I believe the better terms is “convenience” sample.
  • Line 137: was there any basis for choosing n=88? Perhaps a power calculation? Please justify (briefly) the rationale for this sample size.

2.7. Statistical methods: This paragraph needs more detail to explain what analyses were actually carried out. It leads the reader to believe that the Figures of ICC values would map to the different categories or types of equations. However, in the figures in the Results, it does not appear that this was what was done. For instance, in this section, there are twelve different equation types identified. However, in the figures there are variously 12, 17, etc. number or equation categories.

So, for instance, I found the results described from lines 261-296 to be both interesting and very confusing. I could not tell whether you were using the presence of particular variables (e.g., weight, age, etc.) or whether these were equation types. I'm now guessing that you were creating overlapping subsets of studies based on:

  1. the presence of specific variables (e.g., weight, gender)
  2. combinations of variables (weight+gender+age).

But, if so, this still confuses me. Take the example of the "weight" category in Figure 2. All equations with weight as a variable were compared and the ICC=.752. However, in the subset of weight equations that also included gender, age and BMI (weight+gender+age+BMI) the ICC was .534. This is non-intuitive since we'd expect equations with more parameters that theoretically should affect REE to have higher predictive capacity and thus should have higher ICCs. But, even if this is the case (as you describe as “interesting” in the Discussion, it still needs to be clarified to the reader.

Perhaps I am still not understanding the approach to the analyses described in Figure 2 and the following paragraph. Please add more detail to the statistical methods section to clarify your analytic approach.

  • Regarding Table 2: There is a lot going on here but sometimes there appeared to be missing data (for instance, the R^2 value for Carrasco, missing gold standard for Cole & Henry, etc.). If the reason for the missing data is because it wasn’t reported in the original study, then indicating that by using a dash or some such symbol would prevent the reader from thinking that you simply missed the value.
  • Line 202: This is confusing. I thought that studies including patients recovering from cancer were excluded. Did the Kruizenga focus on patients with oncological problems (active cancer?) or merely include these patients?
  • Line 210: You say eleven studies but only reference ten studies.
  • Line 213: you say, “indirect calorimetry was the gold standard most frequently 213 used to measure energy expenditure.” Can you provide the proportion for the reader?
  • Line 230: here you describe age, gender, ethnicity as “demographic characteristics”. However, in Tables 2 and S3 you describe these measures as “anthropometric characteristics” (unfortunately, they are also described as anthropometric characteristics on the web tool as well). This is not the right term. Anthropometric measurements are a series of quantitative measurements of the muscle, bone, and adipose tissue used to assess the composition of the body. See https://www.ncbi.nlm.nih.gov/books/NBK537315/#:~:text=Anthropometric%20measurements%20are%20a%20series,limbs)%2C%20and%20skinfold%20thickness. These should be called “demographic characteristics” throughout.
  • Lines 267-269: You say “Additionally, in males, equations that included laboratory examinations 267 showed a good agreement level (0.94 [(95% CI = 0.87-0.97)]. In examinations of agreement 268 according to gender, females had a higher overall agreement level of 0.67 (95% CI = 269 0.59-0.75), with a narrow CI (Figure 2).” However, I'm not seeing this information in Figure 2. Is this information based on equations that stratify by sex (i.e., separate equations for males and females)? This needs clarification.
  • Figure 3: What does the “measures” category consist of?
  • For Figure 3, it appears that you stratified the type of analyses that you reported in Figure 2 by sex and obesity status. This would be relevant only for equations that themselves stratified by these variables, is that right? If not, it is not clear to me how you did this analysis.
  • Lines 292-296: In the supplemental material associated with the manuscript, the figure numbering begins at S4. So, I do not know what analyses this paragraph is referring to.
  • Section 3.6. Web tool for the practical implementation of equations: I think, but I am not sure, that all the equations that contribute to the outcomes in your example are all equations that include BOTH weight and age (rather than equations that include EITHER weight and age). Basically, I'd like you to confirm whether the algorithm to select the relevant equations uses and AND logic or OR logic. Please make that clear. I don't know that I have a strong feeling about which (AND v OR) would be most appropriate--I rather like the picture of the dispersion which I'd expect you to get with an OR logic since I think it keeps the heterogeneity front and center and avoids falsely precise estimates. But, whatever it is, make it clear to the reader.
  • Lines 426-429: This finding is surprising. I agree that it needs to be tested in a more diverse sample of the elderly. In a relatively homogeneous sample, additional variables may introduce noise though in a more diverse sample, I'd expect that additional health markers may improve agreement. Speculation on my part, of course, but I'm trying to wrap my head around this finding. I wonder if other readers will be scratching their heads.
  • Lines 433-435: I doubt the typical clinician will be able to critically distinguish among the different model results. I like the median of the model results as a sort of "best guess". It made me think a bit of the Rotten Tomatoes app for movies.

Author Response

Dear Ms. Alexandra Draganoiu,

Thank you for giving us the opportunity to submit a revised draft of the manuscript “Resting energy expenditure in the elderly: systematic review and comparison of equations in an experimental population” for publication in the journal Nutrients. We appreciate the time and effort that you and the reviewers dedicated to providing feedback on our manuscript and are grateful for the insightful comments on and valuable improvements to our paper. We have incorporated most of the suggestions made by the reviewers. Please see below, in Italic, for a point-by-point response to the reviewers’ comments and concerns. All page lines number refer to the revised manuscript file with tracked changes.

Dario Gregori, MA, PhD, FACN, FTOS

Unit of Biostatistics, Epidemiology and Public Health, DCTVPH, University of Padova

Phone: +39 049 8275384    --    Fax: +39 02 700445089      --      Cell: +39 347 3518231

Review Report Form

Open Review

English language and style

( ) Extensive editing of English language and style required
( ) Moderate English changes required
(x) English language and style are fine/minor spell check required
( ) I don't feel qualified to judge about the English language and style

Is the work a significant contribution to the field?

Is the work well organized and comprehensively described?

Is the work scientifically sound and not misleading?

Are there appropriate and adequate references to related and previous work?

Is the English used correct and readable?

Comments and Suggestions for Authors

Review of “Resting energy expenditure in the elderly: systematic review 2 and comparison of equations in an experimental population”

First, let me congratulate you on a clever approach to cutting through the fog regarding which equation for BMR/RMR is “best”. The addition of the online tool to translate the research findings to clinical tools was truly creative.

That said, I think that some revisions are needed to make the paper stronger.

First, I do not believe that you carried out a systematic review (which includes an assessment of risk of bias), but rather a scoping review. See https://bmcmedresmethodol.biomedcentral.com/articles/10.1186/s12874-018-0611-x . With the difference between a scoping and a systematic review cleared up, then statements like the one on line 245 about the Cochrane tool are not needed. Thus, I think you need to reframe what you call the first part of the analysis (and, hence, the title of the paper).

We thank the reviewer for his appreciation on our approach.  

We thank the reviewer for pointing out the question related to the risk of bias assessment. We feel that, to some extent, our review is a systematic review, inasmuch it aims to “identify and investigate conflicting results” and follow a structured and pre-defined process. As reported on line 256, in our study, the risk of bias assessment cannot be performed with traditional instruments such as the Cochrane tools, since the studies retrieved in this systematic review were not focused on interventions. Additionally, the risk of bias tools for observational studies (e.g., the Joanna Briggs Institute (JBI) Critical Appraisal Checklist for analytical or prevalence cross-sectional study) cannot be applied to the studies retrieved in our review since the focus of this tools is related to the extent of prevalence or exposure. Furthermore, diagnostic tests risk of bias tools (e.g., the Quality Assessment of Diagnostic Accuracy Studies (QUADAS)-2 tool) were not suitable in our case as they evaluated the performances of the diagnostic test. This may explain why other similar revisions did not provide risk of bias assessment (Gaillard et al., 2007; Schwartz et al., 2012). Other two reviews on the theme have not used traditional risk of bias tools (Frankenfield et al 2005; Madden et al., 2016). The first study created a quality rating checklist and the second instead used a narrative approach evaluating the time lapse between measurement of energy expenditure and variables used in prediction calculations. However, none of these methods were validated. So, we have preferred to use none of these two solutions. We hope our explanation is sufficiently exhaustive with respect to the choice made on the topic of risk of bias assessment.

With regard to the review component of this study:

  • The flow chart is confusing: it appears that there were two title/abstract screens. Please clarify what differentiates the two phases of screening.

We thank the reviewer for pointing this out. The flow chart has been amended accordingly.

  • The next to last box bottom right has been truncated.

We apologize for the mistake: the flow chart has been modified.

  • Please provide a bit more detail regarding the data extraction template. I'm assuming this was a customized template, but was it initially screened for use by the team? Any modifications?

The data extraction template was created based on previous similar works on other populations after reaching a consensus in the working group specialized on the field. The final form has been summarized to present the selected study clearly. The text has been modified as suggested, lines 135-136.

  • On line 127, I think you mean "characteristics of the sample" rather than "population".

We agree with the author. Modified, as suggested.

  • Section 2.5 Data Synthesis: This section really belongs in 2.4 Data Collection as it lists the specific pieces of information extracted from the articles.

We thank the reviewer for the suggestion. The section 2.5 title has been modified as follows: “Data extraction”.

  • Typically, the data synthesis section would describe the meta-analytic models used, methods for imputation or data conversion, subgroup analyses or meta-regressions to explain heterogeneity. Or, if no quantitative synthesis was carried out the use of particular conceptual or theoretical frameworks that guided the qualitative interpretation. That said, it doesn't appear that any particular synthesis was carried out as it seems that the goal of the systematic (actually scoping) review was to identify the equations rather than attempt a traditional synthesis. The lack of typical synthesis for a systematic review is not a flaw, but it should be made clear to the reader who might be expecting something different.

We thank the reviewer for pointing this out. As the reviewer has noticed, this is not a meta-analysis, so there is no need to report any meta-analytical approach. We agree with the reviewer: a data synthesis of the results of selected studies has not been provided. We modified the text accordingly, lines 137-139.

2.6. Data synthesis

The characteristics of each study were summarised in Table 2. Studies were divided according the inclusion of elderly adults in the validation population.

With regard to the observational agreement study:

  • Line 137: you describe your sample as a “casual” sample, though I believe the better terms is “convenience” sample.

We agree with the reviewer. Modified, as suggested, line 141.

  • Line 137: was there any basis for choosing n=88? Perhaps a power calculation? Please justify (briefly) the rationale for this sample size.

We thank the reviewer for pointing this out. The sample used for the predictive equations testing, as reported in line 137 is a “convenient” sample. The 88 subjects were enrolled consecutively among the patients residing in the resting home. So, there are no basis for choosing 88 subjects. On the other hand, the calculation of sample size is not relevant when considering a convenient sample.

2.7. Statistical methods: This paragraph needs more detail to explain what analyses were actually carried out. It leads the reader to believe that the Figures of ICC values would map to the different categories or types of equations. However, in the figures in the Results, it does not appear that this was what was done. For instance, in this section, there are twelve different equation types identified. However, in the figures there are variously 12, 17, etc. number or equation categories.

We thank the reviewer for pointing this out. We agree that groups used for the analysis were not clearly reported. The ICC was computed for 17 group of equations. The paragraph has been modified accordingly, lines 266-271.

“Equations were grouped as follows for the agreement analysis: (a) equations that consider age; (b) equations that consider gender; (c) equations that consider height; (d) equations that consider weight; (e) equations that consider BMI; (f) equations that consider  physical activity; (g) equations that consider more than three variables (three included); (h) equations that include at least one laboratory examination (albumin, glucose level, C reactive protein); (i) equations with at least one measure of circumference (abdominal circumference, hip circumference, wrist circumference) or that include at least one skinfold measure  (chest skinfold, subscapular skinfold); (j) equations including weight and gender; (k) equations with the combination of the variables weight-gender-age; (l) weight-gender-age-height; (m) equations with the combination of the variables weight-gender-age-BMI; and (n) equations with the combination of the variables weight-gender-age-height-BMI equations. For each group was determined the ICC.”

So, for instance, I found the results described from lines 261-296 to be both interesting and very confusing. I could not tell whether you were using the presence of particular variables (e.g., weight, age, etc.) or whether these were equation types. I'm now guessing that you were creating overlapping subsets of studies based on:

  1. the presence of specific variables (e.g., weight, gender)
  2. combinations of variables (weight+gender+age).

We agree with the reviewer, this part has been clarified in lines 266-271.

But, if so, this still confuses me. Take the example of the "weight" category in Figure 2.

We apologize with the reviewer for the lack of clarity. The category “weight” compared all the equations that have at least the variable weight in their structure. Clarified in lines, 266-271.

All equations with weight as a variable were compared and the ICC=.752. However, in the subset of weight equations that also included gender, age and BMI (weight+gender+age+BMI) the ICC was .534. This is non-intuitive since we'd expect equations with more parameters that theoretically should affect REE to have higher predictive capacity and thus should have higher ICCs. But, even if this is the case (as you describe as “interesting” in the Discussion, it still needs to be clarified to the reader.

We thank the reviewer for pointing this out. Clarified in lines 455-462.

“Since equations with more information reduced the agreement among the equations in our sample, we could suggest avoiding the use of equations with that include many variables in their structure, especially for potentially fragile patients, such as those in our sample, for whom all measurements are not usually available. Equations retrieved were usually derived from a specific population, adding variables imply adding coefficient explain the variability of that specific population. This could be the reason why equations with fewer variables showed higher level of agreement in our population.”

Perhaps I am still not understanding the approach to the analyses described in Figure 2 and the following paragraph. Please add more detail to the statistical methods section to clarify your analytic approach.

We agree with the reviewer, this part has been clarified in lines 266-271.

  • Regarding Table 2: There is a lot going on here but sometimes there appeared to be missing data (for instance, the R^2 value for Carrasco, missing gold standard for Cole & Henry, etc.). If the reason for the missing data is because it wasn’t reported in the original study, then indicating that by using a dash or some such symbol would prevent the reader from thinking that you simply missed the value.

We agree with the reviewer, the data not provided in the table were not reported in the original study. Modified Table 2 as suggested.

  • Line 202: This is confusing. I thought that studies including patients recovering from cancer were excluded. Did the Kruizenga focus on patients with oncological problems (active cancer?) or merely include these patients?

We agreed with the reviewer: we have not included equations validated in oncological patients. However, since only a part of the patients included in the study of Kruizenga were oncological patients (29%) (other diseases for the remaining 71%), we have preferred to include it in our review.

  • Line 210: You say eleven studies but only reference ten studies.

There was a mistake in the number of studies reported. The studies were ten. The text has been amended accordingly.

  • Line 213: you say, “indirect calorimetry was the gold standard most frequently 213 used to measure energy expenditure.” Can you provide the proportion for the reader?

We thank the reviewer for this suggestion. The proportion of indirect calorimetry has been provided as suggested, line 226.

  • Line 230: here you describe age, gender, ethnicity as “demographic characteristics”. However, in Tables 2 and S3 you describe these measures as “anthropometric characteristics” (unfortunately, they are also described as anthropometric characteristics on the web tool as well). This is not the right term. Anthropometric measurements are a series of quantitative measurements of the muscle, bone, and adipose tissue used to assess the composition of the body. See https://www.ncbi.nlm.nih.gov/books/NBK537315/#:~:text=Anthropometric%20measurements%20are%20a%20series,limbs)%2C%20and%20skinfold%20thickness. These should be called “demographic characteristics” throughout.

We thank the reviewer for pointing this out. We meant “demographic characteristics”. The tables and the site have been amended accordingly.

  • Lines 267-269: You say “Additionally, in males, equations that included laboratory examinations 267 showed a good agreement level (0.94 [(95% CI = 0.87-0.97)]. In examinations of agreement 268 according to gender, females had a higher overall agreement level of 0.67 (95% CI = 269 0.59-0.75), with a narrow CI (Figure 2).” However, I'm not seeing this information in Figure 2. Is this information based on equations that stratify by sex (i.e., separate equations for males and females)? This needs clarification.

We agreed with the reviewer, the text and figure were not in the correct order. The ICC stratified according to the gender are reported in figure 3. The text and the order of the figures 2 and 3 were modified accordingly.

  • Figure 3: What does the “measures” category consist of?

The category measure “consists of equations with at least one measure of circumference (abdominal circumference, hip circumference, wrist circumference) or that include at least one skinfold measures (chest skin-fold, subscapular skinfold)”.

The text has been clarified, lines 174-178.

  • For Figure 3, it appears that you stratified the type of analyses that you reported in Figure 2 by sex and obesity status. This would be relevant only for equations that themselves stratified by these variables, is that right? If not, it is not clear to me how you did this analysis.

We apologize with reviewer for the lack of clarity. The stratification depicted in figure 3 concerns patients, not equations. This, and the other stratification, are useful to show the level of agreement in a specific subgroup of the sample, as male vs female, regardless of the characteristics of the equation. For example, equations that include BMI in their structure have showed a low agreement both in obese and non-obese patients, as reported in lined 295-297.

  • Lines 292-296: In the supplemental material associated with the manuscript, the figure numbering begins at S4. So, I do not know what analyses this paragraph is referring to.

We apologize with reviewer, there was a mistype on the supplementary figure numbering. Modified the supplementary material and the text accordingly.

Section 3.6. Web tool for the practical implementation of equations: I think, but I am not sure, that all the equations that contribute to the outcomes in your example are all equations that include BOTH weight and age (rather than equations that include EITHER weight and age). Basically, I'd like you to confirm whether the algorithm to select the relevant equations uses and AND logic or OR logic. Please make that clear. I don't know that I have a strong feeling about which (AND v OR) would be most appropriate--I rather like the picture of the dispersion which I'd expect you to get with an OR logic since I think it keeps the heterogeneity front and center and avoids falsely precise estimates. But, whatever it is, make it clear to the reader.

We thank the reviewer for pointing this out. Clarified in the text, lines 344-347. This information is provided also in the app. The logic “AND” was used with categorical variable, “OR” with continuous variable.

" The number of equations resulting from equationer depends on the selected variables. Selecting a choice for categorical variables like, e.g., gender or ethnicity, will result in a lower number of equations estimated. Conversely, setting a value for numerical variables, like, e.g., height or weight, instead will result in a higher number of equations estimated.”

  • Lines 426-429: This finding is surprising. I agree that it needs to be tested in a more diverse sample of the elderly. In a relatively homogeneous sample, additional variables may introduce noise though in a more diverse sample, I'd expect that additional health markers may improve agreement. Speculation on my part, of course, but I'm trying to wrap my head around this finding. I wonder if other readers will be scratching their heads.

We agreed with the reviewer: this result was surprisingly also for us. Still, as reported in our conclusions the results must be confirmed in a comprehensive sample. As far as we have found out, adding more information in the equations is useful to explain the variability among a specific population, but usually this is not always true in populations different from the original.

  • Lines 433-435: I doubt the typical clinician will be able to critically distinguish among the different model results. I like the median of the model results as a sort of "best guess". It made me think a bit of the Rotten Tomatoes app for movies.

We do understand the reviewer’s consideration. Yet the specific aim of this study was not to providing the best equations for the proper patients, but to provide the clinicians  with the instrument to choose among them in his or her own, given the specific circumstances. Further development of this work may allow us to answer which of these equations is best to use. However, a “best guess” as the reviewer effectively depicted, is perhaps more relevant than identifying the “right” equation to use.

Submission Date

14 December 2020

Date of this review

11 Jan 2021 20:43:00

Reviewer 2 Report

A very clinically relevant topic for estimating energy needs in the elderly population. This paper appears to have 3 objectives: to provide the results of a systematic review, to utilize validated REE equations in a sample population of elderly patients, and to create a database of REE equations for clinicians to use.

It seems that this paper would be better described in 3 separate papers, as the data are overwhelming to the reader.  It is unclear what are the hypotheses of this paper? 

Overall suggestion would be to scale back this paper to report 1. the result of the systematic review and at most 2. include data related to using these equations with the population discussed in the paper (n= 87)

Several questions need to be answered:

  1. it is unclear which of the 174 equations found through the systematic review were then validated and narrowed down to 18?  A specific table showing these methods and these specific 18 equations would be helpful.
  2. It is unclear what REE equations were used in the elderly population sample (n=87)?  Did the clinician choose from those equations? If so- this is a separate study.  The most appropriate method would be: for the researchers to utilize all 18 REE equations in the elderly population sample and then compare these with Indirect Calorimetry.  Particularly if the goal is to determine which equations are most accurate for the sample population studied. 
  3. In the absence of being able to do indirect calorimetry, the researchers could still show data related to the use of the 18 validated equations, compare the results to other studies, and plan for further research to validate these equations in this population with indirect calorimetry.
  4. Current data provided is confusing and does not provide a clear conclusion.  Please provide data in a manner that supports a clear conclusion.  It is unclear why the authors conclude that it is best to "not use equations with many measurements" (lines 426-428).  The best way to come to this conclusion is to compare predictive equations with direct measurement (in this case indirect calorimetry).  Please also define "many measurements" (i.e. use only height/weight/age/gender? - and which specific equations the authors suggest?  (i.e. IOM? Harris Benedict?  etc. )
  5. Suggest leaving out data regarding the creation of the clinical/clinician database.  This needs to be tested/studied and reported in another paper.  The current study design is not appropriate for testing the use of a created database- and seems a whole other objective.  In addition, if these data included in the paper is based on clinician decision making, this is a confounder for this study. 

Author Response

Dear Ms. Alexandra Draganoiu,

Thank you for giving us the opportunity to submit a revised draft of the manuscript “Resting energy expenditure in the elderly: systematic review and comparison of equations in an experimental population” for publication in the journal Nutrients. We appreciate the time and effort that you and the reviewers dedicated to providing feedback on our manuscript and are grateful for the insightful comments on and valuable improvements to our paper. We have incorporated most of the suggestions made by the reviewers. Please see below, in Italic, for a point-by-point response to the reviewers’ comments and concerns. All page lines number refer to the revised manuscript file with tracked changes.

Dario Gregori, MA, PhD, FACN, FTOS

Unit of Biostatistics, Epidemiology and Public Health, DCTVPH, University of Padova

Phone: +39 049 8275384    --    Fax: +39 02 700445089      --      Cell: +39 347 3518231

Reviewer 2

Review Report Form

Open Review

English language and style

( ) Extensive editing of English language and style required
( ) Moderate English changes required
(x) English language and style are fine/minor spell check required
( ) I don't feel qualified to judge about the English language and style

Is the work a significant contribution to the field?

Is the work well organized and comprehensively described?

Is the work scientifically sound and not misleading?

Are there appropriate and adequate references to related and previous work?

Is the English used correct and readable?

Comments and Suggestions for Authors

A very clinically relevant topic for estimating energy needs in the elderly population. This paper appears to have 3 objectives: to provide the results of a systematic review, to utilize validated REE equations in a sample population of elderly patients, and to create a database of REE equations for clinicians to use.

It seems that this paper would be better described in 3 separate papers, as the data are overwhelming to the reader.  It is unclear what are the hypotheses of this paper? 

Overall suggestion would be to scale back this paper to report 1. the result of the systematic review and at most 2. include data related to using these equations with the population discussed in the paper (n= 87)

We thank the reviewer for his valuable consideration. While it is true that the article provides a significant number of information, each piece of information is key to try to answer to the question “which equation do I have to use with this patient?”.

The hypothesis of the paper, as reported in lines 71-76, is to provide the clinician with an instrument that helps her to choose, among the numerous equations available in literature, the most suitable for her/his patient.

To this end, we would prefer to maintain the three objectives in the same paper, even if, admittedly, results of this article could benefit from further research. For example, we would like to test these equations on a larger sample and to compare the results with objective measures.

Several questions need to be answered:

  1. it is unclear which of the 174 equations found through the systematic review were then validated and narrowed down to 18?  A specific table showing these methods and these specific 18 equations would be helpful.

We apologize with the reviewer: mistype was present in the text. The equations applied to the sample were 101. All the equations were applied except those for which our sample does not have available data. Table S3 provide the estimated REE for each equation according to gender. The text has been modified accordingly, lines 267-271.

All the equations, except for those that had information that are not available in our sample, were used to compute the REE in our population. For example, the equation of Arciero et al. [38] was not used in our sample since we do not have information regarding the leisure time activity.”

  1. It is unclear what REE equations were used in the elderly population sample (n=87)? 

As answered in the previous point, all the retrieved equations, except those for which we do not have the value in our population in the sample, were applied.

  1. Did the clinician choose from those equations? If so- this is a separate study. 

We thank the reviewer for pointing this out. No clinician selected which equations to use in our sample. If, instead, the question is whether only the equations used in the example are included in the app, then the answer is no. All the retrieved equations are available in the app and the clinician can choose among them.

The most appropriate method would be: for the researchers to utilize all 18 REE equations in the elderly population sample and then compare these with Indirect Calorimetry.  Particularly if the goal is to determine which equations are most accurate for the sample population studied. 

We agree with the reviewer in this consideration, but we feel the aim of this study was slightly different. If the aim was to assess the accuracy of each equation in a specific population, the comparison with an objective measure indeed would be the gold standard. However, our study rather aims to evaluate the agreement among the different equations in the same population.

  1. In the absence of being able to do indirect calorimetry, the researchers could still show data related to the use of the 18 validated equations, compare the results to other studies, and plan for further research to validate these equations in this population with indirect calorimetry.

We agree with the reviewer. In the table S3 the estimated REE for each equation by gender was reported. The study does not aim to validate the retrieved equations, but to show how REE varies according to different equations in the same sample. Hence, we believe our app is a useful instrument to identify, for each case, the most relevant equation, given the characteristics of the specific equation and of the specific patients. When the choice is still challenging, we employed the median value of all the equations used in the app. This approach has showed to reduce the error as reported in lines 413-415. However, as previously reported, we are interested in testing these equations on a larger sample and comparing these results with objective measurement.

  1. Current data provided is confusing and does not provide a clear conclusion. 

Please provide data in a manner that supports a clear conclusion. 

We apologize with the reviewer for the lack of clarity. The methods section has been modified (lines 168-182), the information related to the equations used in our sample were cleared up. The conclusions were also clarified as suggested, lines 455-461.

“This study provides i) a relevant examination of the use of predictive equations for elderly adults, ii) apply the retrieved equations in a convenience sample, and iii) provide a web application to help clinician in the choose of the equations to use. Equations retrieved by this literature review are numerous, consider different variables in their structure, and provide different estimates from one another”

It is unclear why the authors conclude that it is best to "not use equations with many measurements" (lines 426-428). 

We thank the reviewer for pointing this out. Our results showed low agreement among equations that use multiple parameters, as reported in figure 2 and figure 3. Conversely, equations with fewer parameters showed higher level of agreement. This is why we suggest “avoiding the use of equations with that include many variables”. Clarified in line 455-461.

The best way to come to this conclusion is to compare predictive equations with direct measurement (in this case indirect calorimetry).  Please also define "many measurements" (i.e. use only height/weight/age/gender? Clarified in line 455-462. - and which specific equations the authors suggest?  (i.e. IOM? Harris Benedict?  etc. ).

We thank the reviewer for pointing this out. However, our study does not purport to identify the best equation in general, but rather to put clinicians in the position to choose, . We provided instruments to choose among the numerous equations available, for each situation, the most suitable equationamong the numerous equations available.

  1. Suggest leaving out data regarding the creation of the clinical/clinician database.  This needs to be tested/studied and reported in another paper.  The current study design is not appropriate for testing the use of a created database- and seems a whole other objective.  In addition, if these data included in the paper is based on clinician decision making, this is a confounder for this study. 

The application does not collect data, in its simplest form it is a calculator of REE.

Admittedly, the tool will have to be further tested and validated, but we feel this would exceed the subject of this work.

Submission Date

14 December 2020

Date of this review

29 Dec 2020 23:29:38